# Temperature-Controlled Expression of a Recombinant Human-like Collagen I Peptide in *Escherichia coli*

**DOI:** 10.3390/bioengineering10080926

**Published:** 2023-08-04

**Authors:** Wenjie Xie, Qiqi Wu, Zhanpeng Kuang, Jianhang Cong, Qirong Zhang, Yadong Huang, Zhijian Su, Qi Xiang

**Affiliations:** 1Institute of Biomedicine and Guangdong Provincial Key Laboratory of Bioengineering Medicine, Jinan University, Guangzhou 510632, China; xiewj@stu2021.jnu.edu.cn (W.X.); wuqiqi@stu2021.jnu.edu.cn (Q.W.); kuangzhanpeng@foxmail.com (Z.K.); 2022c@stu2022.jnu.edu.cn (J.C.); tydhuang@jnu.edu.cn (Y.H.); 2Biopharmaceutical R&D Center, Jinan University, Guangzhou 510632, China; zhangqq321@outlook.com

**Keywords:** recombinant human-like collagen (rhLCol-I), temperature-controlled plasmid, seamless clone

## Abstract

Collagen is the functional protein of the skin, tendons, ligaments, cartilage, bone, and connective tissue. Due to its extraordinary properties, collagen has a wide range of applications in biomedicine, tissue engineering, food, and cosmetics. In this study, we designed a functional fragment of human type I collagen (rhLCOL-I) and expressed it in *Escherichia coli* (*E. coli*) BL21(DE3) PlysS containing a thermal-induced plasmid, pBV-rhLCOL-I. The results indicated that the optimal expression level of the rhLCOL-I reached 36.3% of the total protein at 42 °C, and expressed in soluble form. In a 7 L fermentation, the yield of purified rhLCOL-I was 1.88 g/L. Interestingly, the plasmid, pBV220-rhLCOL-I, was excellently stable during the fermentation process, even in the absence of antibiotics. Functional analyses indicated that rhLCOL-I had the capacity to promote skin cell migration and adhesion in vitro and in vivo. Taken together, we developed a high-level and low-cost approach to produce collagen fragments suitable for medical applications in *E. coli.*

## 1. Introduction

Collagens are a large family of extracellular matrix proteins and have the capacity to maintain the elasticity and mechanical strength of connective tissue [1]. To date, a total of 28 collagen types have been identified in humans, with the most abundant being types I, II, and III [2]. Type I collagen has the capacity to build skin, bones, tendons, and ligaments. Type II collagen contributes to cartilage formation, and type III collagen helps create muscles and blood vessels [3]. Due to the extraordinary properties of collagen—high tensile strength, fiber orientation, semi-permeability, and low-antigenicity—collagens, especially types I and III, have a wide range of applications in biomedicine, tissue engineering, food, and cosmetics [4,5].

Type I collagen is a fibrous collagen and its most prevalent form [6,7]. It is a trimer consisting mainly of one α2 and two α1 chains [8,9], which interact with cells through several receptor families [10] and self-assemble into intersecting striated fibrils, providing support for cell growth. Type I collagen is widely applied in medical fields such as conventional surgery and ophthalmology, and it is considered the gold standard material in tissue engineering. Lee and colleagues discovered and identified part of a protein that comes from the human COL1A2 gene. This peptide helps dermal fibroblasts grow, type I collagen synthesize and wound healing [11]. Moreover, COLI-decorated nanopores arranged on titanium surfaces seem to essentially direct early inflammatory responses and consequent forms of angio/osteogenesis, leading to favorable osseointegration [12].

Type I collagen is often isolated from the organic connective tissues of distinctive species, such as bovine dermis, fish skin, chicken skin, mammal ligament, and jellyfish tissue. However, the high cost, risk of animal-derived diseases, and some sociocultural beliefs (for example, in Muslim countries and India) limit the potential application and development of this process. To meet the huge application requirements, the most effective solution is to produce these extracellular matrix proteins via recombinant bioengineering technologies [13,14]. Several human collagens have been produced using various recombinant expression systems, including mammalian cells, insect cells, yeast, *Escherichia coli (E. coli)*, and plant cells [15,16,17,18]. Based on its unique molecular characteristics and high molecular weight, it is very challenging to express full-length collagen at high levels in an *E. coli* system [19]. In addition, both *E. coli* and yeast expression systems lack the post-translational enzymatic mechanisms of collagen. Therefore, it is difficult to maintain the molecular stability of the recombinant collagen in the processes of expression, purification and production. In the previous study, we found that the thermal stability of the fragments was higher than that of full-length collagen. The T_m_ value of unhydroxylated collagen is more than 10 °C lower than that of native collagen. 

To achieve better production of recombinant human collagen for various applications, it would be an optimized alternative strategy to express functional fragments of human collagen, according to the results described above, we synthesized a DNA fragment encoding a functional section of type I human collagen and then cloned it into a temperature-controlled expression vector, pBV220 which has been used over the past two decades for protein expression in *E. coli* [20,21,22,23]. 

## 2. Materials and Methods

### 2.1. Construction of Recombinant pBV220-rhLCOL-I Expression Vector

A DNA fragment encoding the functional section of type I human collagen was amplified from the plasmid pET3c-rhLCOL-I (Biopharmaceutical R&D center of Jinan University, China) using rhLCOL-I-SELF-F and rhLCOL-I-SELF-R as the primers (Table 1). A polymerase chain reaction (PCR) reaction was performed using PrimeSTAR HS DNA Polymerase (Takara Biomedical Technology Co., Ltd., Beijing, China) under the following reaction conditions: 98 °C for 10 s, 58 °C for 5 s, and 72 °C for 30 s for a total of 25 cycles. The vector, pBV220 (Beijing Dingguo Changsheng Biotechnology Co., Ltd., Beijing, China), was linearized by PCR with pBV220-SELF-F and pBV220-SELF-R primers under the following reaction conditions: 98 °C for 10 s, 58 °C for 5 s, 72 °C for 4 min. These 2 fragments were examined using agarose gel electrophoresis, purified using a Gel Extraction Kit (Omega Bio-Tek Inc. Guangzhou, China), and ligated via a Gibson Assembly Master Mix kit (New England Biolabs, Beijing, China). The recombinant plasmid, pBV-rhLCol-I, was obtained and then transformed into *E. coli* DH5α (Takara Co., Ltd., Beijing, China). The transformants were screened using PCR with pBV220-TEST-F and pBV220-TEST-R primers, and the sequencing of the plasmids was performed by Shanghai Bioengineering Biotechnology Services Co., Ltd. (Shanghai, China).

### 2.2. Analysis of rhLCOL-I Expression

The validated plasmid was transformed into BL21(DE3) pLysS (Takara Co., Ltd., Beijing, China). Recombinant proteins were inoculated in 5 mL of a fresh Luria-Bertani (LB) medium and incubated overnight in a shaker at 37 °C (220 rpm). The culture was transferred to 5 mL of fresh LB medium (100 μg/mL ampicillin) at a ratio of 1:50 and inoculated at 37 °C. After incubating to an optical thickness (OD600) of 0.6–0.8, rhLCOL-I protein expression was induced in a preheated shaking incubator at 42 °C. The cells were collected after 4 h of incubation by centrifugation at 12,000× *g* for 5 min at 4 °C, then frozen at −80 °C for utilization. The concentrated cells were resuspended in gelling buffer at a proportion of 1:10 (*w/v*) and aerated for 3–5 min. After centrifugation at 12,000× *g* for 10 min at room temperature, the expression of recombinant protein was analyzed using sodium dodecyl sulfate–polyacrylamide gel electrophoresis (SDS-PAGE). Proteins were measured utilizing the ImageJ program, which measures the proportion of target proteins to calculate proteins based on the intensity of the gel bands.

We also constructed an IPTG-induced expression vector as the positive control. Briefly, the DNA fragment coding the functional section of type I human collagen was cloned into the pET-3c vector and then transformed into BL21(DE3) pLysS. After incubation to an OD600 of 0.6–0.8, rhLCOL-I expression was induced with 1 mM IPTG for 4 h.

### 2.3. Solubility Analysis

To assess the solubility of rhLCOL-I, the transformations were inoculated in 50 mL medium and initiated at different temperatures. After 4 h of incubation, the cells were collected, resuspended in PBS at a proportion of 1:10 (*w*/*v*), and broken up by homogenization. After centrifugation at 18,000× *g* for 30 min at 4 °C, the protein fractions of the super-natant and pellet were examined using 12% SDS-PAGE to determine the solubility.

### 2.4. Plasmid Stability

To evaluate plasmid stability during fermentation, a single clone was inoculated in 5 mL of LB medium containing ampicillin. After 14–16 h of cultivation, the culture was transferred to 50 mL fresh LB medium without ampicillin. The culture and induction processes were performed as described above. The samples were taken at the beginning of fermentation and before and after induction. These samples diluted with PBS at 1:100,000 were spread to the LB medium plates with/without ampicillin. The plates were subsequently incubated for 16 h at 37 °C, and the number of emerging clones was calculated.

### 2.5. Fermentation of rhLCOL-I

The seed strain of rhLCOL-I was inoculated into 250 mL Erlenmeyer flask containing 50 mL LB medium with 100 μg/mL ampicillin, shaken at 220 rpm, at 37 °C. When the OD_600_ reached 0.8–1.0, the culture was transferred into 500 mL medium (10 g/L tryptone; 10 g/L yeast extract; 4 g/L NaCl; 1.31 g/L K_2_HPO_4_; 2.5 g/L KH_2_PO_4_; pH, 7.2) and incubated. at 37 °C. With an OD_600_ between 3.0 and 5.0, the culture was further transferred into 7 L of maturation medium (16.3 g/L tryptone; 23 g/L yeast extricates; 12 g/L NaCl; 1.31 g/L K_2_HPO_4_; 2.5 g/L KH_2_PO_4_; 0.6 g/L MgSO_4_; 1.3 × 10^−2^ g/L CaCl_2_; 5 × 10^−3^ g/L vitamin B_1_; pH, 7.2). The agitation and aeration rates of the fermenter were set at 200 rpm and 1.0 (air volume/culture volume/min, vvm), respectively. During fermentation, the dissolved oxygen was kept above 10% of the air saturation, while the pH was maintained between 6.9 and 7.1 by including a 25% (*v*/*v*) ammonia solution and 5 g/L of glucose solution. The maturation temperature was shifted to 42 °C when the wet weight of the cells reached 100 g/L. During the induction phase, the pH was maintained at 7.0 to 7.2 by including 5 g/L of glucose solution. The expression levels of the recombinant proteins and cell density within the cultures were measured at regular intervals. After 4 h of incubation, the cells were collected by centrifugation at 4000 rpm for 60 min at 4 °C, and the cell pellet was stored at −80 °C.

### 2.6. Purification and Identification of rhLCOL-I

Approximately 3 g of cell pellets were resuspended in 30 mL phosphate buffer (20 mM; pH, 6.0). After homogenization, the lysate was centrifuged at 15,000 rpm for 30 min. The supernatant was applied to a Ni-nitrilotriacetate (Ni-NTA) Sepharose column pre-equilibrated with 20 mM PB. After washing with three times the amount of wash buffer (PB with 20 mM imidazole; pH, 6.0), rhLCOL-I was eluted using elution buffer (PB with 500 mM imidazole; pH, 6.0) [13]. The fractions containing the target protein were desalted by employing a Sephadex S-100 column. The purity and concentration of rhLCOL-I were determined utilizing SDS-PAGE and Western blotting analysis, respectively.

### 2.7. Bioactivity Assay of rhLCol-I

Investigation of cell attachment: Crystal violet staining was utilized to assess the adhesion strength of rhLCol-I. Briefly, 100 μL of rhLCol-I (2.5 nmol/mL) was coated on the surface of 96-well plates and incubated overnight at 4 °C. After washing three times with PBS, HaCaT cells (obtained from ATCC, no.: BNCC101683) were inoculated on the plate (1.0 × 10^5^ cells/mL) and incubated at 37 °C for 4 h. Then, the unattached cells were removed by utilizing PBS. Cells that adhered to the wells were immobilized with 2% paraformaldehyde solution for 20 min and stained with 1% crystal violet (Solarbio, Beijing, China) for 20 min.

Cell migration analysis: For the scratch wound test, 5 × 10^5^ cells/well (three duplicates per group) were plated on a 12-well plate and incubated to reach around 95% confluency, after which a scratch was made with a sterile pipette tip. Then, the cells were treated in DMEM medium containing 0.5% FBS supplemented with rhLCol-I (2.5 nmol/mL). The cells were imaged at 48 h and 72 h. The closure range of the wound was calculated as follows: relocation range (%) = (A_0_ − A_n_)/A_0_ × 100, where A_0_ represents the introductory zone of the wound range and A_n_ represents the remaining region of the wound at the measurement point.

### 2.8. Minimally Invasive Skin Recovery of rhLCol-I

Male Sprague–Dawley (SD) rats (6–8 weeks, 190–220 g) were obtained from the Guangdong Medical Laboratory Animal Center (Guangzhou, China; certification number: 44007200069979). The rats were housed in the animal facilities of the Jinan School of Medicine under controlled light conditions (12 h light, 12 h dark) with free access to food and water. All animal tests were approved by the Animal Ethics Committee, Jinan University, Guangzhou (2020826-11). The animals were anesthetized with 1% sodium pentobarbital solution (Sigma, Osterode am Harz, Germany) via intraperitoneal injection. After dorsal skin debridement, four circular (12 mm diameter) abrasion scald wounds were created by applying mechanical friction and scalding at 70 °C for 15 s. After the creation of the trauma surface, the rats were randomly divided into four groups. Trauma changes were measured regularly, and the time of crusting, debridement, redness, and swelling resolution were recorded. ImageJ software was used to perform semi-quantitative analyss of the trauma areas. Rats were euthanized at 3, 7, and 14 days postoperatively. The skin tissues surrounding the wound site were isolated and fixed in 4% neutral-buffered paraformaldehyde, which was paraffin-embedded, sectioned at 8.0μm thickness, and stained with H&E and Masson trichrome.

### 2.9. Statistical Analysis

The data are displayed as mean ± standard deviation (SD) based on at least three independent tests. Data analysis was conducted utilizing GraphPad Prism 9 (GraphPad Inc., Jolla, CA, USA). In the case of different groups, one-way ANOVA was utilized, followed by Tukey’s HSD comparison test. The statistical significance of the results was set at *p* < 0.05.

## 3. Results

### 3.1. Plasmid Construction and Expression of rhLCOL-I

The DNA fragment encoding rhLCOL-I was cloned into the expression vector pBV220 using the Gibson assembly method, and the accuracy of the rhLCOL-I sequence was confirmed via DNA sequencing. A schematic diagram of rhLCOL-I cloned into pBV220 is shown in Figure 1a. Nucleic acid electrophoresis analyses of the pBV220 vector, pBV220- rhLCOL-I, and rhLCOL-I PCR fragments are shown in Figure 1b,c. The expression vector pBV220-rhCOL-I was designed to transform *E. coli* BL21(DE3) pLysS. The expression of a ~26 kDa protein, compared to the anticipated estimate, was initiated by incubating at 42 °C for 4 h. For shake-flask fermentation, the optimal expression reached 36.3% of the total protein compared to the IPTG-induced control (Figure 2a). Interestingly, the expression of rhLCOL-I could be activated when the temperature was higher than 39 °C (Figure 2b). Moreover, both recombinant collagen fragments, induced by IPTG and temperature, were detectable by Western blotting (Figure 2c).

### 3.2. Plasmid and Thermal Stability of rhLCOL-I

It is well known that the stability of non-integrating plasmids, maintained by antibiotics, is closely related to the expression level of bioengineered proteins. In this study, we found that the lack of antibiotics during fermentation did not lead to a significant loss of the recombinant pBV220 plasmid. There was no significant difference between the samples cultured with/without antibiotics (Figure 3a). Importantly, an inductive process at high temperatures also did not reduce the stability of the recombinant plasmid (Figure 3b).

High temperature is the one of critical factors resulting in the formation of inclusion bodies in recombinant proteins. After induction at 42 °C, the cells were collected, resuspended, and sonicated. The result of SDA-PAGE shows that rhLCOL-I was mainly expressed in soluble form, indicating that rhLCOL-I has excellent thermal stability (Figure 2c).

### 3.3. Fermentation, Purification, and Identification of rhLCOL-I

To optimize rhLCOL-I expression in scale-up fermentation, the cells harboring pBV220-rhLCOL-I were induced at 42 °C when the average wet weight of cell pellets reached 100 g/L. After 4 h, the biomass yielded 130.71 ± 11.47 g/L. The cells were collected, sonicated, and centrifuged at 4 °C for 30 min. Subsequently, the lysate was applied to conduct Ni-NTA Sepharose and Sephadex S-100 column chromatography. The results of SDS-PAGE and HPLC indicated that the purity of rhLCOL-I was over 90% (Figure 1d and Figure 2e). The average yield of purified rhLCOL-I was 1.88 g/L (*w*/*v*), significantly higher than that of the IPTG-induced system (approximately 0.52 g/L).

### 3.4. rhLCOL-I Promoted HaCaT Cell Migration and Adhesion

To investigate the functions of recombinant human collagen fragments, the confluent HaCaT monolayer cells were scratched and treated with 0 or 2.5 nM/mL rhLCOL-I. The results showed that rhLCOL-I had no obvious ability to promote cell proliferation (Figure 4a). However, both recombinant collagen fragments, from different *E. coli* expression vectors, were able to significantly decrease the wound gap after treatment at 48 h. The wound gap further reduced to ~8.4% compared to the starting time (or 0 h) in the control group at 72 h. Crystal violet staining can be used to analyze the number of adherent cells present in cell culture plates. The presence of rhLCOL-I significantly increased the number of cells, which was 2.3-fold higher than that of the control (Figure 4b). Collectively, the above findings strongly suggest that rhLCOL-I has a significant capacity to promote skin-derived cell migration and adhesion.

### 3.5. rhLCOL-I Improved the Wound-Healing Process

To illustrate the treatment of the rhLCOL-I hydrogel, an in vivo wound-recuperating measurement was performed by making circular wounds on the back skin of rats. The creatures were divided into four treatment groups: PBS solution (control), vehicle (mock), pET3c-rhLCOL-I (positive control), and pBV220-rhLCOL-I. For visualization, the dynamic wound-recuperating process was followed and outlined in a schematic graph, as shown in Figure 5. The results demonstrate that the wound regions in all four groups steadily decreased. Compared with the control and mock groups, the rhLCOL-I groups showed significant enhancements in the wound-healing process at an early stage, in addition to reduced aggravation.

A histological investigation of the wounds was performed to evaluate the skin solution. The control group showed greater invasion by inflammatory cells, more pronounced hemorrhagic centers, and looser connective tissue than the other groups. In addition, the regenerated epithelial tissues within the rhLCOL-I treatment groups were thicker and more contoured than those of the control and mock groups (Figure 6). These results illustrate that rhLCOL-I contributed to advancing wound healing.

## 4. Discussion

The basic physicochemical properties and biological activities of collagen are high tensile strength, biodegradability, low antigenic activity, low irritation, and low cytotoxicity [24]. These characteristics make collagen an ideal biomedical material for various clinical applications. At present, the development of the collagen industry is mainly subject to high costs and low production. In China, the market price of collagen suitable for injection or implantation is over 200,000 CYN/kg. However, over the past few decades, recombinant collagens have been created as models to study the biophysical characteristics of collagen and different protein–collagen interactions. Thus, the production of recombinant collagen with simple manufacturing and high efficiency is very important.

Nowadays, scientists are progressively designing microbial consortia for various applications, including the bioproduction of solutions, biofuels, and biomaterials from renewable carbon sources. Biotechnologists have planned multi-collagen [25] protein platforms to spatially organize frameworks within a couple of nanometers for effective biosynthetic pathways [26]. These scaffolded bioactive complexes can be fastened to the surfaces of manufactured materials, such as permeable agarose globules, cellulose particles, etc. In addition to central hydroxylation issues, the exceedingly repetitive structure and frequent Gly and Pro residues in collagen may also cause expression problems [27]. There is another issue regarding how to utilize high-level recombinant expression frameworks for the generation of stable triple-helical human collagens. Myllyharju et al. reported [28] that P. pastoris may be effective for advancements in custom-made expression frameworks aimed at the specific generation of non-secreted triple-helical and secreted single-chain collagen fragments of various lengths for particular purposes. Although there are more and more new technologies emerging for recombinant expression of all kinds of collagens, such as types III, V, and VII, their commercial production is also limited by the high repetition frequency of sequences and low expression levels [29,30,31].

In our previous research, we found that purified rhLCOL-I formed a fibrous precipitate when the temperature decreased below 4 °C. When the temperature increased, it would dissolve into hydrogels. Moreover, the thermal stability of type I collagen was consistently below 28 °C and increased at higher temperatures (33 °C) [32]. Therefore, we speculated that the functional fragment of type I human collagen also has this potential ability. As expected, it showed excellent thermal stability at 42 °C and was expressed in a fully soluble form. More interestingly, during fermentation, the pBV-220 vector was more stable in BL21(DE3) PlysS than in DH5α in the absence of antibiotics. The mechanisms involved deserve further investigation.

Like full-length collagen, rhLCOL-I maintained the capacity to promote skin cell migration and adhesion in vitro and in vivo. However, it lacked a stimulatory effect on skin cell proliferation. Therefore, compared with the mock group, rhLCOL-I did not significantly increase the process of wound healing. This flaw can be improved by simultaneously adding growth factors or chemicals. In addition, an alternative option is to modify rhLCOL-I with functional peptides.

In summary, we developed a high-level and low-cost approach to produce collagen fragments fit for medical applications related to *E. coli*. It is very suitable for production on a mass scale.

## Figures and Tables

**Figure 1 bioengineering-10-00926-f001:**
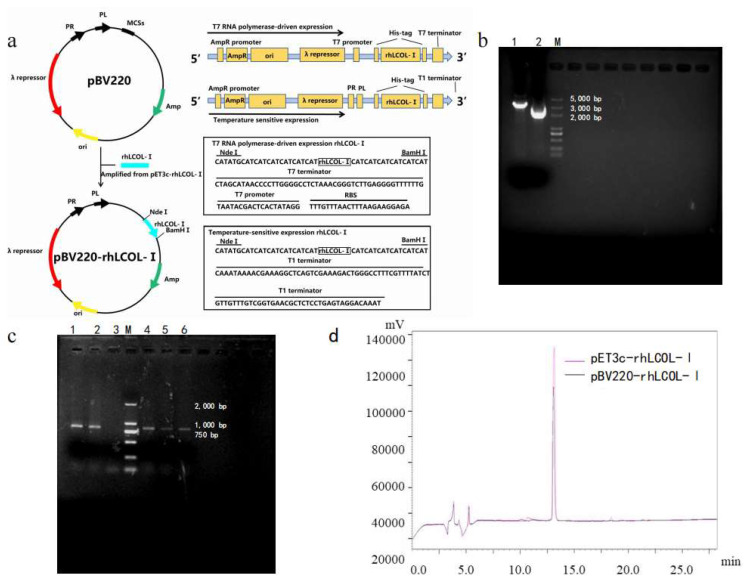
Construction schematic of recombinant pBV220- rhLCOL-I plasmid. (**a**) Construction map of the recombinant plasmid. Construction map of the plasmid pBV220-rhLCOL-I. The rhLCOL-I gene was cloned into the plasmid pBV220 to construct the plasmid pBV220-rhLCOL-I. PR and PL indicate the lambda PR promoter and PL promoter, respectively. MCSs—multiple cloning sites; ori—origin. (**b**) Nucleic acid electrophoresis analysis of the pBV220 vector and pBV220-rhLCOL-I. Lane 1: the vector of pBV220-rhLCOL-I; lane 2: the vector of pBV220; lane M: DL5000 marker. (**c**) Nucleic acid electrophoresis analysis of rhLCOL-I PCR fragment. Lane M: DL2000 marker; lanes 1–6: rhLCOL-I PCR fragment. (**d**) Analysis of purified rhLCOL-I from different vectors using SEC-HPLC.

**Figure 2 bioengineering-10-00926-f002:**
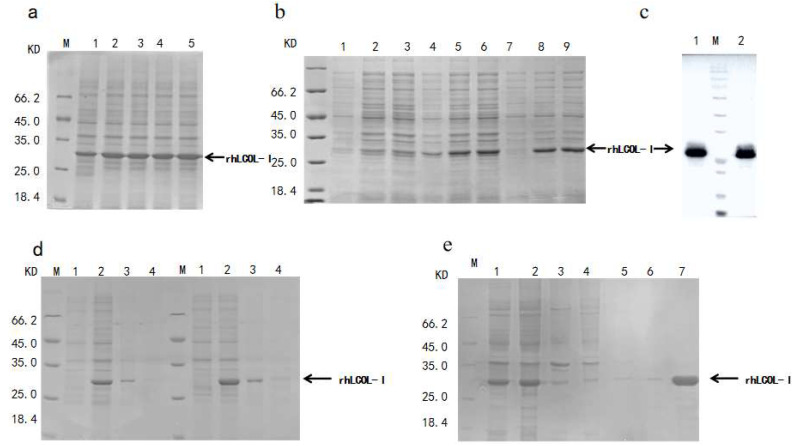
Expression of rhLCOL-I. (**a**) SDS-PAGE analysis of rhLCOL-I expression. Lane M: molecular-weight markers; lane 1: rhLCOL-I expression using pET3c(+) vector; lanes 2–5: rhLCOL-I expression using pBV220 vector. (**b**) SDS-PAGE analysis of the best expression temperature of rhLCOL-I in *E. coli* using a water bath. Lane M: molecular-weight markers; lanes 1–3 and 4–6: rhLCOL-I expression using pBV220 vector at 38 °C and 39 °C, respectively, repeated for three groups; lane 7: rhLCOL-I expression using pBV220 vector at 37 °C; lanes 8–9: rhLCOL-I expression using pBV220 vector at 42 °C, repeated for two groups. (**c**) Western blotting analysis of rhLCOL-I on different vectors. Lane 1: rhLCOL-I expression using pET3c(+) vector; lane M: molecular-weight markers; lane 2: rhLCOL-I expression using pBV220 vector. (**d**) SDS-PAGE analysis of the solubility of rhLCOL-I. Lane M: molecular-weight markers; lane 1: rhLCOL-I expression before induction; lane 2: rhLCOL-I expression after induction; lane 3: the expression of rhLCOL-I in the supernatants; lane 4: precipitate after the breakage of bacteria. (**e**) Expression and purification of rhLCOL-I were checked on SDS–PAGE. Lane M: molecular-weight markers; lane 1: rhLCOL-I expression using pBV220 vector after induction; lane 2: the expression of rhLCOL-I in the supernatants; lanes 3–4: flowthrough; lanes 5–7: supernatants eluted with 200 mM and 500 mM imidazole.

**Figure 3 bioengineering-10-00926-f003:**
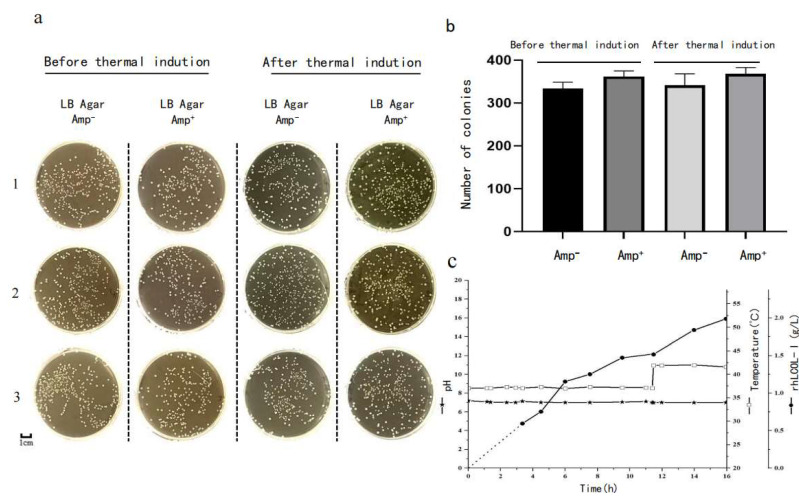
The recombinant plasmid’s stability and a schematic diagram of rhLCOL-I production during fermentation. (**a**) The recombinant plasmid stability was assessed by comparing the clone numbers. The transformats were collected from the fermented broth before or after thermal induction and then inoculated to the LB plates with or without ampicillin, respectively. The experiments of inoculation were repeated 3 times (marked with 1, 2, and 3, respectively). Scale bar = 1 cm. (**b**) The graph shows the colony numbers of the samples collected from before and after the thermal induction. (**c**) Changes in pH, temperature, and protein yield of rhLCOL-I during fermentation. The inoculation experiments were repeated 3 times (marked with 1, 2, and 3, respectively). Scale bar = 1 cm.

**Figure 4 bioengineering-10-00926-f004:**
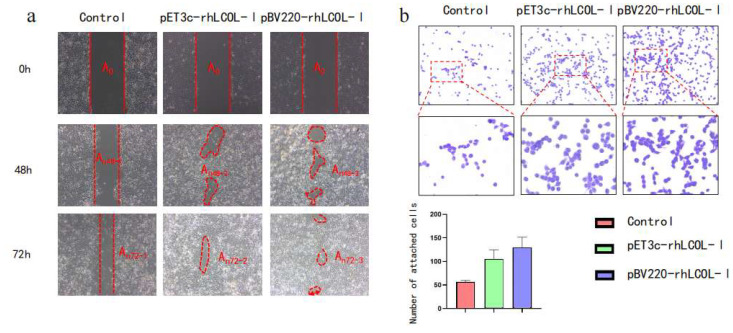
HeCat cell migration and adhesion. (**a**) At 0, 48, and 72 h after wound formation, in vitro migration experiments were carried out on a carrier, pEt3c(+), and pBV220, respectively, to obtain images. The red dotted line indicates the scratch boundary. (**b**) Crystal-violet-stained and magnified images showing cell adhesion. The graph shows the quantitative analysis of attached HaCat cells cultured on the control, pET3c-rhLCOL-I, and pBV220-rhLCOL-I groups. Scale bar = 100 μm. The red dotted line indicates the enlarged dotted line box.

**Figure 5 bioengineering-10-00926-f005:**
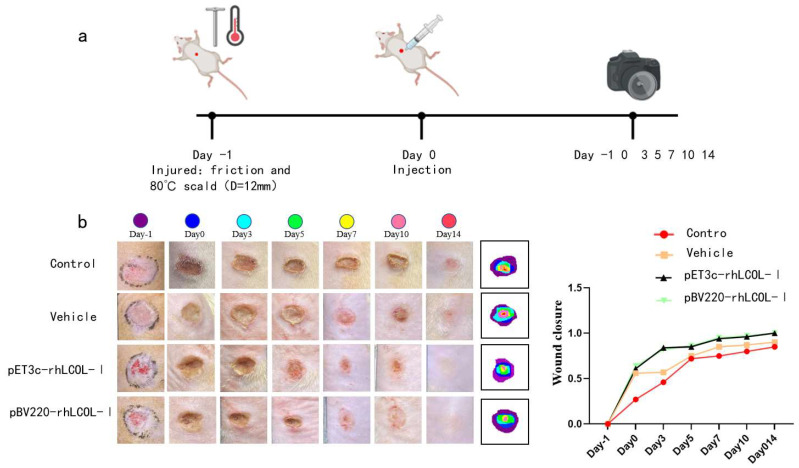
Minimally invasive skin repair test on SD rats. (**a**) Wound areas for wounds treated with vehicles and rhCOL-I at −1, 0, 3, 5, 7, 10, and 14 days post-surgery. n = 6, means ± SD. (**b**) Wounds were photographed on days −1, 0, 3, 5, 7, 10, and 14. Traces of wound closure were found at 14 days post-surgery (scale bar = 12 mm).

**Figure 6 bioengineering-10-00926-f006:**
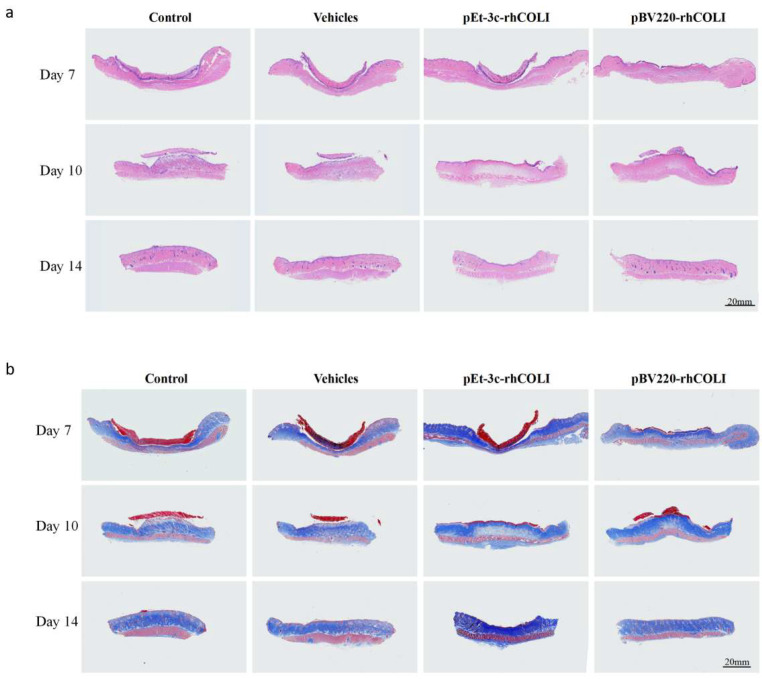
Skin wound tissue sections of rats on days 7, 10, and 14. (**a**) H&E staining of wound tissues. (**b**) Masson’s trichrome (MT) staining of wound tissues.

**Table 1 bioengineering-10-00926-t001:** The list of PCR primers.

Primer Name	Primer Sequences (5′–3′)
pBV220-SELF-F	CTAGGATCCGTCGACCTGCAGCCAAGCTTC
pBV220-SELF-R	GATGATGCATATGGAATTCCTCCTTA
rhLCOL-I-SELF-F	GGAGGAATTCCATATATGCATCATCATCATCATCATA CTAGTGGCGAACGTG
rhLCOL-I-SELF-R	TGGCTGCAGGTCGACGGATCCTTAATGATGATGATGATGA TGACTA
pBV220-TEST-F	CTGAGCACATCAGCAGGAC
pBV220-TEST-R	ACAGAAGCTTGGCTGCAGGT

## Data Availability

Not applicable.

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
