# Peer review of "Temperature-Controlled Expression of a Recombinant Human-like Collagen I Peptide in Escherichia coli"

_bioengineering, 2023, doi:10.3390/bioengineering10080926_

Round 1

Reviewer 1 Report

Review Comments

Bioengineering MS Title: “Pilot-scale expression of a recombinant human-like collagen I peptide (rhLCol-I) from E. coli”

This MS deals with the production of rhLCol-I from cloning, expression through purification and the analysis of molecular and functional characteristics of the purified protein. The content is scientifically sound, suitable to the Bioengineering journal, and thus should be of interest to the journal readers. However, to improve its quality, extensive revision is deemed necessary. My suggestions for the revision are shown below.

1. “Pilot-scale” in the title can be ambiguous depending on how it is defined.

2. [p. 21] Should be corrected to p < 0.05?

3. In numerous places, English expressions are incorrect, awkward, and grammatically wrong. Thorough English editing is required by a native speaker.

4. Detailed information is missing such as amino acid sequence and/or MW on the rhLCol-I peptide/protein to be expressed.

4. [Introduction] This section is occupied by quite a few sentences and references on the traditional sources and usage of the collagens. Some of them can be deleted to focus more on the need for recombinant production.

5. [Line #152, 154, 157, 158, 170; “according to ….”] The meanings of theses phrases are uncertain and need to be clarified.   

6. [Result] This section needs to be strengthened by adding the significance/inference of the data.

7. [Line #270, Fig 1C; #278, Fig. IIa, Figure; #281, Fig. Ie, #294, Fig. 3A] These do not appear in the figures and/or are mismatched with the text.   

8. Table 1 and 2 are not referred to in the text.

9. [Discussion] Much of the sentences are inappropriate for discussion and can be relocated to Introduction. It needs major revision to focus on the meaning and significance of the data and findings of this study.

The objective, experimental procedure and content, and the resulting data of this MS are acceptable; however, it needs to be reorganized and revised for publication.

[END]

See the Comments to Authors.

Reviewer 2 Report

Report

The present research bioengineering-2421634 titled: “Pilot-scale expression of a recombinant human-like collage I peptide (rhLCol-I) from Escherichia coliwas aimed at heterologous expression and improvement of the yield of a recombinant human-like collage I peptide (rhLCol-I) from Escherichia coli: The topic is very interesting from the environmental aspect, however; there are some comments and suggestions that should be considered and fulfilled and these as follows:

Comments

1.     The introduction section to too long with many unnecessary citations. For example, from L38 to L48 the authors described the advantage of collagen I as compared to the other types. Then in L54, the authors mentioned “Type I collagen is an important biomaterial.[18][19], with the citation of two review articles.  So there are a lot of redundancies and unnecessary reference citations. The author should thoroughly revise the introduction section and make it shorter, concise, and point and focuses mainly on the relevant literature particularly those regarding the recombinant expression of collagen I in different host and 2.     L82, a reference is needed for the pBV220 vector otherwise it should be transferred to the Methods section and described in detail and insert the vector map to reveal the multiple cloning site, Promoter, and antibiotic-resistant marker. The construction by the Institute of Virus Research, Academy of Preventive Medicine, China is not enough in this case. The authors mentioned reference 36 for the criteria of this vector however, the reference is not correct and irrelevant to the mentioned information” CIts has promoter repressor activity at 30°C and loses promoter repressor activity by inactivation at 42°C”. There are many irrelevant reference citations, therefore, I recommend that the authors must revise each reference cited for the respective mentioned information and avoid excessive citations of review articles particularly for experimental or biological activities or vectors instead citation of original research articles is important in this case. 3.     All the names of microorganisms should be italicized in the whole manuscript. For example, E. coli should be E. coli 4.     Abbreviations should be first described at the first mention and then used consistently in the whole manuscript (examples, amino acids abbreviations should be written in full at the first mention. The whole manuscript should be accordingly thoroughly revised. 5.     Citation of the reference should be added at the end of the sentence and before full stop”.) and not after. The whole manuscript should be revised accordingly. 6.     In the Methods, the authors should include the sequences, source, and/or reference of the used primers used in this study and underline the restricted sites that were created for cloning 7.     In the results, I recommend including a figure of agarose gel electrophoresis showing the PCR amplicon, empty plasmid, and recombined plasmid (before and after double digestion) to confirm the size of the DNA insert and verify successful coning. 8.     SDS-PAGE and Western blot analysis need citation of the relevant reference. Also, more details are needed for Western analysis and the types of antibodies used to determine the expressed soluble protein. 9.     A source or a reference of E. coli DH5α and E. coli BL32(DE3) should be provided in the method section. 10.  In the result the author did not provide the histopathological examination of the wound tissues stained with H&E and Masson trichrome. The picture of the wound per day is not supportive and instead, a table of average wound diameter and standard deviations should be included for each animal group. The conclusion section should be more elaborated to include the novelty of the obtained results and new findings of this study together with highlighting future perspectives. Therefore, and for the above-mentioned remarks, I advised a major revision of the respective manuscript in its current state taking into consideration the above comments and recommendations before being considered for publication

overall good with minor typing error

Round 2

Reviewer 2 Report

The authors have properly responded to all of the addressed queries and suggestions and now the manuscript becomes suitable for publication.

well-written and well presented with very minor grammatic mistakes

Author Response

Manuscript Number: Manuscript ID: bioengineering-2421634

Pilot-scale expression of a recombinant human-like collage I peptide (rhLCol-I) from Escherichia coli

Dear editors and Reviewers,

Thank you for your letter and for the reviewers’ comments concerning our manuscript entitled “Pilot-scale expression of a recombinant human-like collage I peptide (rhLCol-I) from Escherichia coli”  Manuscript ID: bioengineering-2421634). Those comments are all valuable and very helpful for revising and improving our paper. 

  • At first, we checkedall references carefully. They are all relevant to the contents of the manuscript.
  • All revisions to the manuscript are marked in red.
  • We send the manuscript to a native speaker for editing and corrections onweb of https://susy.mdpi.com/user/manuscripts/resubmit/6c6af6eedaaaa2fb8649d71b371781c6. The English proofreading certification is in the attachment.

We appreciate for Editors/Reviewers’ warm work earnestly, and hope that the correction will meet with approval.

Once again, thank you very much for your comments and suggestions.

Wenjie Xie
